# The Recent Advances in the Utility of Microbial Lipases: A Review

**DOI:** 10.3390/microorganisms11020510

**Published:** 2023-02-17

**Authors:** Sajid Ali, Sumera Afzal Khan, Muhammad Hamayun, In-Jung Lee

**Affiliations:** 1Department of Horticulture and Life Science, Yeungnam University, Gyeongsan 38541, Republic of Korea; 2Centre of Biotechnology and Microbiology, University of Peshawar, Peshawar 25120, Pakistan; 3Department of Botany, Garden Campus, Abdul Wali Khan University Mardan, Mardan 23200, Pakistan; 4Department of Applied Biosciences, Kyungpook National University, Daegu 41566, Republic of Korea

**Keywords:** microbial lipase, purification, industrial application, lipase engineering, biocatalysis

## Abstract

Lipases are versatile biocatalysts and are used in different bioconversion reactions. Microbial lipases are currently attracting a great amount of attention due to the rapid advancement of enzyme technology and its practical application in a variety of industrial processes. The current review provides updated information on the different sources of microbial lipases, such as fungi, bacteria, and yeast, their classical and modern purification techniques, including precipitation and chromatographic separation, the immunopurification technique, the reversed micellar system, aqueous two-phase system (ATPS), aqueous two-phase flotation (ATPF), and the use of microbial lipases in different industries, e.g., the food, textile, leather, cosmetics, paper, and detergent industries. Furthermore, the article provides a critical analysis of lipase-producing microbes, distinguished from the previously published reviews, and illustrates the use of lipases in biosensors, biodiesel production, and tea processing, and their role in bioremediation and racemization.

## 1. Introduction

Enzymes are biocatalysts present in all living organisms. The uniqueness of the biological activity of enzymes is found in the precision with which they act, utilizing minimal energy to catalyze a specific reaction. Enzymes play a pivotal role in all stages of metabolism and biochemical reactions. Some enzymes are of special interest as they can be used as catalysts in various biochemical processes and, hence, have a wide application. Lipases are a subclass of esterases and perform an important role in the digestion, transport, and processing of lipids in most living organisms. They are versatile and enable a range of bioconversion reactions (hydrolysis, alcoholysis, acidolysis, aminolysis, esterification, and interesterification) in unicellular and multicellular organisms. Lipases are essential for the bioconversion of triacylglycerols (TAG) from one organism into another organism and within organisms.

Similarly, lipases possess a distinctive feature of acting at an interface between the aqueous and non-aqueous phases, which is unique and distinguishes them from all other esterases. Other unique properties of lipases include the following: specificity, pH dependency, temperature, catalytic activity in organic solvents, and their nontoxic nature. The most desirable features of lipases also include their ability to process all types (mono-, di-, and triglycerides) of glycerides as well as free fatty acids in transesterification, their excellent activity in nonaqueous media, a low level of product inhibition, and tolerance to temperature and pH fluctuation. Additionally, lipases are stable in organic solvents and active without cofactors [1,2,3]. Lipases are found in all living beings and are utilized for their normal functioning. The use of microbial (fungi, yeast, and bacteria) lipases has received considerable attention over the past several decades [4,5].

In recent times, microbial biotechnologists have shifted their attention to the commercial use of lipases of microbial origin, and numerous microbial strains have been screened and characterized for enzyme production. The microbes most frequently used for lipase production include *Penicillium* sp., *Candida rugose*, *Aspergillus niger*, *Rhizopus* sp., and *Pseudomonas* sp., [6,7,8,9,10,11,12,13,14,15,16,17]. A plethora of microorganisms belonging to different genera of fungi, bacteria, and yeast have been isolated from different environmental conditions and screened. These microorganisms are potential sources of lipases [18,19]. Some recent studies on microorganisms and their potential to produce lipases are mentioned in Table 1. Lipases of microbial origin have revealed distinct physiochemical and biological characteristics that favor their use as prominent biocatalysts and have proved to be competent alternatives to classical organic techniques in the selective transformation of complex molecules in different industries. Lipases are commonly used in the food industry and play an effective role in the manufacture of a variety of food products (baked food, juices, and fermented foods). Similarly, lipases are extensively used in industrial cleaners, leather processing, cosmetics, paper, and detergent industries, while other applications of lipases in biosensors, biodiesel production, biomedical applications, pesticides, and bioremediation are of great importance.

Based on their total sales, lipases are considered one of the largest groups of enzymes. The international lipase market is anticipating sales greater than USD 797.7 million by 2025 [19]. The global microbial lipase market size was USD 349.8 million in 2019 and is expected to maintain a compound annual growth rate (CAGR) of 5.2% in the forecast period of 2020 to 2025, reaching USD 428.6 million by 2025. According to some recent reports, the microbial lipase market stands out as one of the most proactive industry verticals because most of the enzymes currently used in different industrial processes are of microbial origin. All the industries mentioned earlier require purified lipases for the biocatalytic production of their products.

The conventional fermentation and purification techniques used for the manufacture of microbial lipases exhibit various limitations as most lipases are extracellular and require fermentation followed by filtration or centrifugation. The major limitations of conventional purification approaches include low productivity and prolonged process times. Interestingly, researchers have presented new approaches and adopted the latest technologies for the purification of lipases of microbial origin. Some of the recent technologies include immunopurification techniques, an aqueous two-phase system, aqueous two-phase flotation, and reversed micellar systems. The main objective is the economical production of large quantities of lipase for industrial use.

The significance of lipases, specifically microbial lipases, can be seen from the substantial number of research and review articles published in the last decade. All these studies have contributed to intellectual growth and provided technical information regarding fermentation, purification, and applications of lipases. The novelty of the present article is based on reporting the diverse aspects of microbial lipases and discussing the research gaps. Thus, in the present review, we have conducted a comprehensive analysis of the available data on the production of lipases from different microbial sources, their applications in different areas, some modern approaches for the purification of lipases, and recent trends and targets adopted in lipase engineering.

**Table 1 microorganisms-11-00510-t001:** Microbial sources of lipases from diverse environments.

Microbial Source	Isolated from	Applications	Reference
		Fungal Species	
*Penicillium simplicissimum*	Food samples	Olive oil as an inductor to increase the production of lipase; can be used in various food industries	[20]
*Yarrowia lipolytica IMUFRJ50682*	Isolated from an estuary in Guanabara Bay	Potential application in the hydrolysis of fish oil to further produce polyunsaturated fatty acids in a suitable process	[21]
*Penicillium roqueforti*	Organic debris	Uses in ethyl oleate synthesis	[22]
*Aspergillus niger*	Soil samples	Various industries where low-temperature enzymatic reactions are required	[8]
*Arbuscular mycorrhizal fungi* (AMF)	Soil samples	AMF can boost plant nutrient absorption and resilience to a variety of abiotic stresses	[23]
*Agaricus bisporus*	Wood-rotting samples	The Laccase enzyme is responsible for the color change of the *Boletus* genus mushrooms when they come into contact with air.	[24]
*Acremonium implicatum*	Buds of *Panax notoginseng*	Detects antimicrobial activity against phytopathogens	[25]
*Chaetomium* spp.	Marine green alga	Has the ability to synthesize a variety of bioactive chemicals that are lethal to leukemia cell lines	[26]
*Eupenicillium* spp.	Soft coral-derived fungus	An important resource for finding active natural products.	[27]
*Fusarium moniliforme*, *Paecilomyces* sp.	Root tip of plantain	This is the organism most used for the production of gibberellic acid through submerged fermentation	[28]
*Pestalotiopsis* spp.	Chinese mangrove plant Rhizophora stylosa	Used to find chromones, coumarins, isocoumarin derivatives, cytosporones, lactones, alkaloids, and terpenoids, among other bioactive natural compounds	[29]
*Daedalea quercina*	Wood-rotting samples	Capable of producing antioxidative and anti-inflammatory compounds	[30]
*Penicillium* spp.	Soil and water samples	Well recognized for its use in the manufacture and ripening of blue-veined cheese	[31]
*Talaromyces* spp.	Soil samples	Used as food preservatives and coloring for several hundred years and still used in Asia for the production of certain items	[32]
*Beauveria bassiana*	Air samples	Used for the control of insect pests	[33]
*Chaetomium globosum*	Marine green alga	Used to produce a wide range of bioactive substances; these compounds showed considerable cytotoxicity against leukemia cell lines	[26]
*Alternaria* spp.	*Catharanthus roseus*	Produces alternariol 51, alternariol 5-O-sulfate 52, and alternariol 5-O-methyl, among other polyketides; these chemicals have an anticancer effect on a variety of cancer cells, including oral human epidermal carcinoma (KB) cells	[34]
*Leptosphaeria maculans*	Leaf of canola/rapeseed seedlings	Causes blackleg disease in canola/rapeseed	[35]
*Colletotrichum falcatum*	Sugarcane	Causes red rot disease in sugarcane	[36]
*Phoma* spp.	Crop plants	Commonly occurs on crop plants that are economically important and causes devastating plant diseases	[37]
*Penicillium roqueforti*	Organic debris	Uses in ethyl oleate synthesis	[22]
		Yeast Species	
*Meyerozyma guilliermondii*	Cheese whey	Capable source of feed lipase used in cheese whey	[38]
*Wickerhamomyces anomalus*	Grapes and wines	Can increase certain characteristics of wine	[39]
*Kluyveromyces marxianus*	Traditional dairy products	Acts as an efficient cell factory to produce various metabolites including lipase	[40]
*Saccharomyces cerevisiae*	Organic materials	Has the ability to produce wine with the best combination of chemical and aromatic characteristics	[41]
*Pichia pastoris*	Exudates of a chestnut tree	Has the ability to produce recombinant protein	[42]
*Rhodotorula glutinis*	Isolated in Antarctica	Has the capability of producing lipids and growing quickly; can be used in various industrial processes	[43]
*Trichosporon* spp.	Lignocellulosic biomass	Has the potential to produce ethanol under anaerobic conditions in synthetic media and sweet sorghum	[44]
*Brettanomyces* spp.	Brewing products	Has a huge potential to add new flavors to the craft beer repertoire	[45]
*Cryptococcus neoformans*	Bird samples	Is a leading cause of morbidity and mortality in immunocompromised individuals, such as patients suffering from HIV/AIDS	[46]
*Geotrichum candidum*	Tibet kefir milk	Can remove more than 99% of Pb^2+^ ions in water at low concentrations and a maximum of 325.68 mg lead/g from dry biomass	[47]
*Hansenula anomala*	Fermented soybean	Has the capability of excellent lipase production	[48]
		Bacterial Species	
*Enterobacter cloacae*	Acidic conditions	Enantioselective esterification potential for pharmaceutical applications	[49]
*Serratia nematodiphilia*	Paper and pulp effluent	Paper and pulp effluent in deinking process in making of paper and increasing brightness, intensity, and the pulping rate	[50]
*Exiguobacterium* sp. strain (AMBL-20)	Glacial water samples	Exhibits potential in bio-detergent formulation	[51]
*Escherichia coli*	Acidic conditions	Has the ability to grow in moderate acidic conditions	[52]
*Acidithiobacillus ferrooxidans*	Acidophilic and chemolithotrophic sulfur- and iron-oxidizing bacterium	Widely used in the bioleaching process for extracting metals	[53]
*Leptospirillum* spp.	Acidic environments	Has the ability to survive at low pH and can be used in different industrial processes	[54]
*Roseobacter* spp.	Marine environment	Alphaproteo bacteria with diverse metabolic and regulatory capabilities	[55]
*Bacillus* spp.	Abattoir soil	Bacterial isolate showed positive lipase activity at the end of 24 h incubation	[56]
*Bacillus amyloliquefaciens*	Environmental samples	Media was optimized with cheap agro-industrial wastes and investigated for enhanced production of lipases in solid-state fermentation	[57]
*Anoxybacillus gonensis* UF7	Isolated from a hot spring	*Anoxybacillus* lipase was used for the first time as catalyst for biodiesel production	[58]

## 2. Sources of Microbial Lipases

Lipases perform a pivotal role in the digestion, transportation, and processing of dietary lipids in most living organisms. Animals, plants, and microorganisms produce lipases. For several reasons, microbial lipases are preferred for commercial use. Microbial lipases are more easily produced in larger quantities for commercial use compared to plant and animal lipases [59,60]. Microbes from diverse environmental conditions (contaminated soils, hot springs, and glaciers) are sources of lipases with unique characteristics [61,62,63]. Similarly, microorganisms from aquatic and endophytic environments exhibit a higher capability for the synthesis of lipases [64,65,66]. In commercial applications, lipases from fungi, yeast, and bacteria are commonly used and different species are explored for the synthesis and purification of lipases. Additionally, microbes are considered the most suitable sources of temperature-resistant and solvent-tolerant lipases, and the use of extremophilic lipases is highly recommended due to their easy production, purification, greater stability, and desired molecular modifications [67]. Microbes produce lipases both extracellularly and intracellularly. Extracellular lipases are exploited extensively because they can be easily separated from the culture media and purified [60,68]. Thus, different species of fungi, yeast, and bacteria from diverse environmental conditions are used to produce lipases and have attracted a great deal of attention. Various biotechnological applications have been extensively utilized for enhancing the availability of lipases for their use in commercial bioprocesses and biotransformation [59,62,69].

### 2.1. Lipases from Fungi

Initially, porcine and human pancreases were exploited for lipases. However, currently, the principal sources of lipases and phospholipases are filamentous fungi, followed by yeast and bacteria [62,63]. Indeed, lipases from fungi are presently receiving considerable attention. Fungi are widely regarded as one of the best lipase sources because of their substrate specificity and stability under various chemical and physical conditions. Additionally, lipases from fungal sources are mostly extracellular in nature and significantly reduce the cost of downstream processing, making this source preferable over other microbes, such as bacteria, that produce lipase intracellularly. Since the 1950s, lipases from fungal sources have been systematically studied, and current industrial processing favors the use of fungal lipases in different fermenters. Similarly, fungal species producing lipase are routinely utilized for cleaning spillages of oil and dairy products, and the prolific role of lipase-producing fungi has also been widely reported in the process of bioremediation. In particular, filamentous fungi are more active in solid-state fermenters and exhibit greater efficiencies compared to submerged fermentation. There have been reports of the production of lipases from several different filamentous fungi including *Aspergillus niger*, *Aspergillus oryza*, *Rhizopus nodosus*, *Penicillium roquefortii*, *Penicillium abeanum*, *Penicillium chrysogenum*, *Trichoderma lanuginosus*, *Fusarium oxysporum*, *Alternaria*, etc. [70,71,72], and these are the primary sources utilized to produce commercial lipases.

### 2.2. Lipases from Yeast

Yeast has been used in different food processing procedures and other industries since ancient times. Yeast lipases are easy to handle and grow in comparison to filamentous fungi and bacteria [73,74]. Most of the yeast lipases are produced extracellularly and more than 50% of the reported yeast-producing lipases are in the form of various isozymes. The yeast *Candida rugosa* is the most commonly used source of lipases for commercial purposes [73]. The lipases of *Candida rugosa* exhibit high activity both in hydrolysis and synthesis and are used in several different processes. Knob et al. [38] isolated yeasts from slaughterhouse refrigerators and oil mill effluents and screened them for lipolytic activities. Among 25 yeast samples, the *Meyerozyma guilliermondii* (L1) strain was selected as a good lipase producer and robust results (6.7-fold increase in lipase production) were obtained when *M. guilliermondii* was grown in 2.0% cheese whey for 24 h. Similarly, yeasts such as *Pichia pastoris* and *Saccharomyces cerevisiae* are used as popular expression systems to produce recombinant proteins because yeast cells have significant advantages compared to bacterial cells, e.g., posttranslational modification, speed of growth, secretory expression, and easy genetic recombination [42]. Thus, researchers are greatly interested in using yeast biological expression systems to produce recombinant proteins, such as lipases, for industrial purposes.

### 2.3. Lipases from Bacteria

Various bacterial strains from different environments, such as seawater, glaciers, hydrothermal vents, industrial effluents, sediments, permafrost, and soil, have exhibited the capability of producing lipase enzymes [51]. The most important bacterial genera that have been investigated for the production of lipases include *Bacillus*, *Pseudomonas burkholderia*, *Streptomyces*, *Arthrobacter*, and *Archomobacter*. The lipases from these bacterial sources are not commonly used for food processing and are mainly used in detergent formulation and biodiesel production. Temperature is the main factor that influences the growth and physiology of microbes used to produce desired products such as lipases. Bacterial strains belonging to the genus *Exiguobacterium* can survive under harsh environmental conditions (−12 to 55 °C) and the lipases from *Exiguobacterium* are functional over a wide range of pH and temperatures, which makes them suitable candidates for use in different industrial and agricultural processes [75]. Similarly, lipases from *Pseudomonas glumae*, *Pseudomonas mendocina*, *Pseudomona salcaligens*, *Pseudomonas fluorescens*, *Bacillus thermoscatenulatus*, and *Burkholderia cepacia* are used in the biofilm degradation, detergent, and biodiesel industries [62,76,77,78]. More recently, Zhao et al. [79] isolated and identified a new lipolytic bacteria, *Staphylococcus caprae* (NCU S6), from sewage and evaluated its growth characteristics for the production of a novel lipase for industrial and biotechnological applications. Lipolytic bacteria are categorized into distinct families for lipase synthesis based on their genetic sequences, biochemical capabilities, and their habitats.

The fundamental advantages of microbial lipases are due to their multifunctional uses and the convenience they afford for bulk production. Additionally, microbial lipases are of great importance because of their ease of purification, molecular modification, activity, stability, and capability for continuous production compared to lipases from animal and plant sources.

## 3. Purification Strategies of Microbial Lipases

Lipase purification is a meticulous technique that requires great care to retain the bioactive state of the lipase. Lipases have been purified from various microbial origins to a level of homogeneity by using different purification techniques. These purification strategies can be broadly divided into two categories: (A) classical purification techniques and (B) modern purification techniques. The classical techniques are usually non-specific, laborious, and multi-step, and the purity level achieved is not adequate. On the other hand, modern purification techniques are stress-free, specific, large-scale, and can reach high purity levels. Improvements in purification strategies have paved the way for a variety of choices in the design of highly specialized purification techniques for different microbial lipases. The purification of the lipases to a level of homogeneity helped in the identification sequence of the amino acids and their 3D structure, which led to a better understanding of their unique characteristics when used in different reactions.

### 3.1. Classical Purification Techniques

#### Precipitation and Chromatographic Separation

Initially, filtration or centrifugation procedures were employed for the purification of lipases from a culture broth. Microbial lipases are mostly extracellular and fermentation is followed by the separation of the cells from the culture media. During the purification process, precipitation is the initial step which is followed by chromatographic separation techniques. Almost all purification schemes use a precipitation step in which ammonium sulfate, ethanol, acetone, or hydrochloric acid are most commonly used. According to the literature available on the subject, the precipitation step has been attempted in more than 80% of the purification schemes, in which ammonium sulfate is used twice (60%) as often as ethanol or acetone (30%). Precipitation is followed by a combination of various chromatographic techniques such as affinity chromatography and gel filtration [80].

A recent study by Bharathi et al. [81] revealed the use of a cell-free supernatant as a crude enzyme for the precipitation and dialysis process, wherein ammonium sulfate fractions from 20 to 80% (*w*/*v*) were used. It was concluded that an ammonium sulfate fraction from 40–60% (*w*/*v*) showed a higher lipase precipitation activity compared to the other fractions used in the study. The precipitation enzymes were then purified using a dialysis membrane. While lipase activity is boosted with the increasing concentrations of the ammonium sulfate solution, the above study also revealed that increasing the saturation of ammonium sulfate above 80% (*w*/*v*) halted the activity. More interestingly, precipitation procedures have high average yields (87%) compared to other techniques [80]. In addition, halophilic lipase purification using three steps involving precipitation with ammonium sulfate and performing ion-exchange chromatography twice was reported [82].

Generally, a single chromatographic procedure is insufficient and various combinations of chromatographic steps are used to achieve the required level of purity. The methods used most frequently are ion exchange, affinity, and adsorption chromatographies [80]. The selection of the appropriate chromatographic method depends on the lipase preparation and purification scheme. The ion exchangers routinely used are the diethyl amino ethyl (DEAE) group in anion exchange and the carboxy methyl (CM) group in cation exchange. This is followed by the gel filtration purification method [80,83,84]. Modern purification techniques are employed for lipase purification to improve lipase productivity and reduce processing time. Developments in purification procedures have widened the choices available for selection when designing specialized purification schemes for microbial lipases.

### 3.2. Modern Purification Techniques

The advantages of modern purification techniques are primarily due to their high yields and shorter process times as compared to classical purification techniques. Currently, the industrial sector is seeking lipase purification techniques that are economical, speedy, and offer a high yield. Below, we provide examples of modern purification techniques which have been covered in recently published review articles. However, here we focus on the applied aspects of these techniques and their commercial values.

#### 3.2.1. Immunopurification

The application of affinity chromatography for the purification of a target protein by using an antibody–antigen system is known as immunopurification (Figure 1). This novel technique is also known as immunoaffinity chromatography, which can be applied for the robust separation of specific proteins. In immunopurification, affinity-purified polyclonal and monoclonal antibodies are used for the separation of target proteins.

Initially, this was regarded as an expensive technique. However, new developments in antibody mass production have paved the way for the deployment of immunopurification at the industrial level by using different monoclonal antibodies [85,86]. A study by Rahimi et al. [87] revealed two monoclonal antibodies (MoAbs), BF11 and VNH9, that immobilize lipase obtained from *Candida rugosa*, wherein both antibodies of the IgG1 isotype recognize specific antigenic determinants shared by different *C. rugosa* lipase isoforms. The residual activity of lipase was found to be 99% (for BF11) and 92% (in the case of VNH9) after forming an immune complex [19,87]. This showed the potential of monoclonal antibodies for application in the purification of microbial lipases.

#### 3.2.2. Reverse Micellar Systems (RMS)

Reversed micelles are associated with the idea of a microreactor where the enzyme can be protected from the detrimental effects of the solvent. The structure of a reverse micelle consists of an aqueous microdomain facing the polar head of the surfactants that surround this core and interact with the bulk organic solvent, which is non-polar, through hydrophobic chains [21,88]. The reverse micellar system (RMS) comprises two essential steps known as forward and reverse extraction (Figure 2). In the liquid–liquid separation process of forward extraction, lipase is transported from an aqueous medium to an organic phase where the enzyme is encapsulated in reversed micelles and protected from denaturation upon contact with the organic solvent, while in reverse extraction, the lipase does not remain in the reverse micelles for a long time but is simply transported into an aqueous media without any contaminant. Recently, surfactants are routinely used in RMS because they enhance the solubility of organic compounds and reduce surface tension [89,90]. In the context of industrial enzyme uses, the question of whether and how lipases and surfactants interact frequently arises. The active conformation of some lipases has been demonstrated to be stabilized by non-ionic surfactants that mimic the lipid substrate, although anionic surfactants’ catalytic interactions have not been thoroughly studied [91]. With a hydrophilic head group attached to a hydrophobic tail, surfactant structures resemble the natural substrates of lipases. Due to this resemblance, surfactants are also used to capture the open conformation of the lipases during crystallization studies. For instance, the open form of a member of the thermoalkalophilic lipase family’s structure was recorded when non-ionic surfactants were present. More recently, Shehata et al. [91] reported that no experimentally obtained ionic surfactant-captured lipase structures exist, despite the fact that a number of studies have shown that ionic surfactants stabilize lipases [88].

In RMS, surfactants expand the interfacial area of the solvent which augments micelle formation and enzyme recovery from the system. This role of the surfactants and the formation of the micelle pave the way for robust downstream processing and a higher degree of enzyme purification [90]. Gaikaiwari et al. [89] applied both the classical approach and reverse micellar approach for the purification of lipase from *Pseudomonas* and concluded that the RMS approach resulted in a 15-fold higher purification with 80% recovery in 45 min. Conversely, the conventional system resulted in 52% enzyme recovery and required 30–40 h. The RMS exhibited the fastest and most economical procedure for lipase purification from bacterial sources. Similarly, the role of lipase from a thermophilic fungus, *Thermomyces lanuginosus,* was reported by Fernandes et al. [92] in the hydrolysis and synthesis reaction in an RMS which produced a higher yield (200 U/mg) in a short period of time.

#### 3.2.3. Aqueous Two-Phase System (ATPS)

The ATPS technique has achieved prominence in purification technology and the application of this technique has resulted in rapid developments in the separation and purification of biomolecules. ATPS is a single-step technique with high selectivity and an easy scale-up process. ATPS is one of the more refined systems for the purification of biomolecules such as DNA, RNA, proteins, and other cellular components. The unique characteristics of this system include a continuous mode of operation, low toxicity of the phase-forming chemicals, and high biocompatibility. The ATPS is a liquid-to-liquid extraction strategy that requires lower energy and provides a relatively quick separation of the target biomolecules into two distinct phases (Figure 3). The purification of the biomolecules is based on the incompatibility of the two phases and the ATPS exploits this unique characteristic for extraction and purification.

In the ATPS, polymers and salt are used for phase separation. The most common polymer-based ATPSs use dextran and polyethylene glycol (PEG), while polymer–salt ATPSs include the PEG–magnesium sulfate ATPS and PEG–potassium phosphate ATPS [93]. Souza et al. [94] used an ATPS based on tetrahydrofuran (THF) and potassium phosphate buffer for the purification of lipases from *Burkholderia cepacia*, *Candida antarctica*, and *Aspergillus niger,* confirming the potential of the THF-based ATPS for lipase purification [95]. The ATPS thus demonstrated its utility as an attractive alternative technique that can meet the greater demands of industrial processes. This technique is also beneficial in terms of economics and environmental sustainability.

#### 3.2.4. Aqueous Two-Phase Flotation (ATPF)

Bi et al. [96] proposed a novel technique to separate and concentrate Penicillin G from a fermentation broth. In this technique, separation by solvent sublation was performed in an aqueous two-phase system and was termed aqueous two-phase flotation (ATPF). Broadly, the ATPF technique is a combination of ATPS and solvent sublation. ATPF is a more effective technique for separating and concentrating biomolecules from an aqueous phase with the reduced consumption of organic solvents. In ATPF, nitrogen gas is passed through the bottom of an enzyme-rich solution which adsorbs the surface-active compounds. Subsequently, the nitrogen bubbles dissolve in the top polymer phase (Figure 4). ATPF is an economical and environmentally friendly technique for the separation and purification of lipases from fermentation broths. However, further studies are required to investigate the optimum rate of flow of nitrogen gas, bubble size, and flotation rate constant for the maximum purification of lipases and other biomolecules.

The use of phase-partitioning chemicals in a conventional ATPS makes it expensive and limits recycling. On the other hand, novel thermoseparating polymers such as ethylene oxide (EO: 50%) and propylene oxide (PO: 50%) (EOPO) can be used [96,97]. This allows the system to be recycled and achieves higher productivity. Both ATPS and ATPF require a two-step procedure. Initially, the target protein is accumulated and extracted by the top phase of EOPO. This top phase of the initial ATPF results in a secondary phase which separates at a lower critical solution temperature. The top aqueous (water) phase contains the target proteins or biomolecules, while the bottom aqueous phase contains concentrated EOPO solution, which can be reused by recycling for further purification of the target protein. ATPF was used for the direct recovery of lipase derived from *Burkholderia cepacia* (ST8) from a fermentation broth. The ATPF was composed of EOPO copolymer and ammonium sulfate, wherein the recovery of up to 75% of EOPO and successful purification of the bacterial lipase in a single downstream processing step was reported [97].

## 4. Uses of Microbial Lipases in Various Industrial Processes

Lipases are used in several industrial processes. Lipases derived from microbial sources are of great importance in different industries as compared to lipases derived from plant or animal sources. Because microbes can be grown easily, have a simple genome (such as bacteria), and can be genetically modified to deliver the desired traits, a continuous supply of the desired products can be achieved with high yields and easy and economical processing. The prolific effects associated with microbial lipases motivated their market growth and applications in various industrial processes. Microbial lipases are extensively used in different industries such as food, textile, leather, cosmetics, paper, detergent, pharmaceutical industries, etc. Below, we discuss some of the industrial applications of microbial lipases.

### 4.1. Food Industry

Lipases have become an integral part of the modern food industry. The varied applications of lipases in the food industry have enhanced the global demand for microbial lipases. Microbial lipases are of great importance in the tailoring of vegetable oils, improving the flavor of cheese, production of linear meat, flavor development of other dairy products, and processing of foods containing fat [98].

Fats and oils are the main constituents of our daily food. The tailoring of fats and oils is one of the main objectives of the food processing industry which demands novel economical and green technologies. Lipases modify the properties of lipids by tailoring the location of fatty acid chains in glycerides, resulting in the conversion of a less desirable lipid into a high-value fat [99]. Similarly, lipases catalyze the hydrolysis, esterification, and interesterification of oil and fats obtained from different sources. Low-quality oils could be upgraded to synthesize nutritionally important low-calorie triacylglycerols (TAGs) [98]. Due to their specific enzymatic reactions under moderate conditions, lipases are likely to occupy a prominent place in the edible oil industry. Similarly, lipases modify flavors and fragrances by synthesizing the esters of short-chain fatty acids and alcohols [98,99]. Lipases from *Pseudomonas* spp., *Bacillus* spp., *Penicillium* spp., *Rhizopus* spp., and *Mucor* spp. are used in different processes in the dairy products industry. Microbial lipases are widely used for the hydrolysis of fats in milk, which modifies the length of the fatty acid chain, boosts the flavor of cheese, and, in particular, aids in the formation of soft cheese [100,101]. Recently, individual microbial lipase or mixtures of microbial lipases have been used for the preparation of good quality cheese [102]. Moreover, lipases of microbial origin have been frequently used in refining the flavor of rice, enhancing the fragrance of apple wine, and modifying soybean milk [103,104]. Additionally, lipases are used for biolipolysis, in which the level of fat in meat is decreased by adding lipases to produce leaner meat [103,105].

In the black tea processing industry, lipases are used for the induction of the hydrolytic cleavage of membrane lipids and the initiation of the synthesis of volatile products with a unique flavor. The quality of black tea is based on the dehydration and enzymatic fermentation of tea leaves. Ramarethinam et al. [106] reported the use of fungal lipases from *Rhizomucor miehei* to enhance the aroma of black tea. The lipases augmented the level of polyunsaturated fatty acid (PUF) by reducing the total lipid content [106].

### 4.2. Textile Industry

In the modern textile industry, microbial lipases are used for degreasing purposes, often called fat trimming. In degreasing, oil/grease on the surface of the textile raw materials is removed to enhance the performance and staining process [107]. Similarly, the quality of wool is improved by removing the fatty acids present on the surface of the wool by using anhydrous alkaline lipase and limiting the cracks in the denim abrasion system [108]. The application of microbial lipases in the surface modification of synthetic fibers and the desizing of cotton fabrics individually or in combination with other enzymes, such as protease or zylanases, is attracting the attention of researchers associated with the textile industry all over the world [109]. Additionally, the use of lipases increases the dying ability of fabrics with lower surface damage and weight loss [110,111]. Microbial lipases from *Geobacillus* spp., *Pseudomonas* spp., *Candida* spp., *Aspergillus* spp., and *Streptomyces* spp. are routinely used in the textile industry. Selvam et al. [112] showed that the fragrant ester synthesized by the *S. variabilis* NGP 3 lipase mediated an enzymatic reaction and concluded that these enzymes demonstrated the maximum production of ester compared to other sources. They also reported that the synthesized esters were imparted to the knitted fabric by exhaustion and the microencapsulation method [112].

More recently, AbouTalib et al. [113] used thermally stable lipase enzymes produced from thermophilic *Bacillus aryabhattai* B8W22 in the bio-scouring of wool. They immobilized the enzymes on sericin-based discs to enhance their stability and make them reusable. Their report revealed the utilization of immobilized thermophilic lipases for the first time in the benign bio-scouring of wool fiber to augment its dye ability with acid, basic, and reactive dyes [113]. Similarly, fungal lipases are the commercial enzymes most in demand, and the industrial sector requires continuous improvements in their production processes and efficiencies together with cost reductions [114].

### 4.3. Leather Industry

The leather industry contributes significantly to the economic prosperity of countries that produce high-quality leather. However, this sector causes serious environmental pollution because of its excessive use of chemicals. In leather manufacturing, the fat between the grain surface and the corium (the inner layer of the skin) is normally removed through dry cleaning with strong organic solvents. However, the process of dry cleaning is expensive and can cause damage to the skin. Various types of detergents are also used for cleaning fats. This is considered a safer technique but is not as effective as dry cleaning. On the other hand, lipases are soluble enzymes that act on insoluble TAGs. The use of lipases in the leather industry is gaining acceptance as a safer, faster, and more economical technique. Additionally, lipases improve filling and dye penetration to give the leather a uniform appearance [115].

Aloulou et al. [116] reported that LIP2 lipases (YLLIP2) are produced by the yeast *Yarrowia lipolytica* at an industrial scale of up to 3 g/L of the culture medium. Moujehed et al. [115] used YLLIP2 in the degreasing process for sheep skin instead of the harmful chemicals previously used in this process. They also reported that 6 mg of lipase/kg of raw skin resulted in successful degreasing within 15 min at a pH of 8 at 30 °C [115,117]. Similarly, a study by Ben Rejeb et al. [118] reported the optimization of enzymatic degreasing of sheep leather by using lipases from *Aspergillus niger*, *Rhizopus oryzae*, *Penicillium roquefortii*, and *Candida rugosa* in a fractional experimental design with four different parameters, viz., the source of enzyme, processing stage, amount of lipase, and duration of degreasing. Their results revealed that the enzymatic degreasing efficiency was higher compared to the conventional process and that lipases from *Rhizopus oryzae* exhibited significant hydrolysis. They also concluded that lipase degreasing improved leather quality and reduced the use of chemicals and surfactants.

Fat removal through detergents or any other chemicals leads to a rise in the effluent physiochemical parameters that are associated with several other problems, such as the requirement of a high amount of organic solvent, environmental pollution, and low quality of leather. Thus, the use of lipases in the degreasing process is considered to be faster, cleaner, and more robust compared to conventional chemical methods. The use of lipases also maintains the quality of leather when compared with conventional chemical treatment.

### 4.4. Cosmetics Industry

Generally, commercial personal care items are regarded as cosmetics. In 2008, the market for cosmetic chemicals in Europe, the US, and Japan was estimated to be about USD 6.8 billion, excluding natural soaps, solvents, and fragrance materials. Recently, based on their applications, cosmetics have been categorized into different categories, including skin care, perfumes, hair care, toiletries, decorative cosmetics, mouthwash, toothpaste, and bath additives [119]. Lipases are considered the most important enzymes in the production of different types of cosmetics because they are important as both active ingredients in the formulation of a cosmetic material and as biocatalysts in the synthesis of specific chemicals used in cosmetics. Active lipases are mainly used in cosmetics for superficial cleaning [120], anti-cellulite treatment, and overall body slimming or removal of dirt and small flakes of dead corneous skin. Similarly, lipases find other applications in the cleansing of the nose, hair care, and beauty masks [119]. Appropriate preparation of lipases is an important task for their application as active ingredients in cosmetic formulation or as biocatalysts in the synthesis of chemicals. Presently, only immobilized enzymes have gained industrial acceptance in the cosmetic industry, while commercial processes are based on free enzymes, as the final product needs to be free of residual enzymes.

Lipases are commonly encapsulated in nanoparticles for use as active ingredients in cosmetics [121]. Particles used for lipase encapsulation comprise a core material in which the lipases are mixed with an oily dissipating medium. This is stabilized with aluminum di- or tristearate. Agar is used as a peripheral material [119]. Fujiwara and Nakahara [122] reported that hollow spheres of inorganic silica could be promising materials for lipase encapsulation. A study by Miguez et al. [123] discussed the optimized enzymatic synthesis of a cosmetic ester (decyloleate) catalyzed by a homemade biocatalyst prepared through the physical adsorption of lipase (from *Thermomyces lanuginosus*) on amino-functionalized rice husk silica. Their results showed 87% ester conversion in a solvent-free system, and the biocatalyst retained its activity even after 8 successive batches. Various techniques are used for the application of lipases in cosmetic formulations based on the application procedure that is acknowledged for a particular interest in industrial processes. Lipases play a minor role as functional cosmetics and a major role as biocatalysts for the industrial production of esters, fragrance compounds, and other active ingredients.

### 4.5. Paper Industry

Lipases are used for the modification of raw starch in the paper industry. Lipases are also used for the removal of the pitch, which is an insoluble component of wood and mostly contains triglycerides and waxes. Pitch adversely affects the process of paper manufacturing, while the microbial lipases used for the removal of pitch do not affect either the environment or the quality of paper [124,125]. Initially, Shehata [91] and his coworkers at Nippon Paper Industries (Japan) developed a pitch control method using lipases that catalyze the hydrolysis of triglycerides. After confirming the prolific effects of lipases in reducing the level of pitch, their protocol was universally adopted and flourished in the papermaking industry. The factors that need to be controlled for optimizing production using lipases in this industry include enzyme concentration, temperature, reaction duration, and the rate of continuous mixing. Lipase from *Candida rugosa* is used in the hydrolysis of 90% of the triglycerides or waxes [1,102]. Similarly, lipase from *Candida cylindracea* exhibited good results in the hydrolysis of triglycerides in the extracts of fresh birch pulp and hydrolyzed 30% of the esterified lipids [125,126].

### 4.6. Detergent Industry

Lipases are used in detergents to enhance their cleaning ability. Lipase-based detergents have better cleaning properties compared to synthetic detergents, as they are active at low temperatures, can be used in small quantities, do not lose their activity after removing stains, and are environmentally friendly [127]. The formulation of lipase-based detergents for the removal of fat soil from fabrics includes an ionic or nonionic surfactant and lipases. The use of liquid laundry detergents which use enzymes in an encapsulated form is increasing. Cold-active lipase detergents are used for cold washing to reduce the wear and tear of the fabric and to reduce energy consumption [128,129]. Lipases from microbial sources are widely used in the detergent industry, and alkaline yeast lipases are preferred compared to fungal and bacterial lipases [127].

Maharana and Singh [130] reported the isolation of psychrotolerant yeast *Rhodotorula* sp. Y-23 to produce lipase using a plate assay followed by submerged fermentation, and described an optimized process for the production of the maximum amount of lipase using palmolein oil (5% *v*/*v*) at 15 °C. They also reported potential inducers (galactose, potassium nitrate [KNO_3_], and manganese chloride [MnCl_2_]) for lipase production, evaluated the tolerance capacity of the enzyme, and concluded that lipase was compatible with commercially available detergents. The addition of lipases from psychrotolerant yeast *Rhodotorula* sp. Y-23 to different detergents augments lipid degradation, making it a potential candidate for use in the detergent industry [130].

Lipases from bacterial sources have been classified into eight different families, based on their specific properties, because there is a variation in the optimal pH and temperature between these families [131]. Normally, bacterial lipases have a neutral or alkaline optimal pH, while lipases with acidic optimal pH values are produced by *Pseudomonas fluorescens* and *Pseudomonas cepacia*. Psychrophilic lipases are also reported to be produced from *P. fluorescens* with optimal activity at 20 °C [132,133,134]. Phukon et al. [12] reported a novel *Pseudomonas helmanticensis* HS6 for the production and characterization of lipase for application in the detergent industry. They reported an 18.78-fold enhanced production of purified lipase which could be active over a wide range of temperatures (5–80 °C) and pH (6–9) while showing optimum activity at 50 °C at a pH of 7. Additionally, the lipase from *P. helmanticensis* HS6 retained residual activity of 40–80% after 3 hours of incubation with commercial detergents, which suggests its suitability for applications in the detergent industry.

## 5. Role of Lipases in Biosensors

The use of lipases in biosensors or biological assays is an application of growing interest. Lipases are frequently used because of their wide-range substrate specificity and greater commercial availability. In biosensors, lipases can be used for two different purposes. They can be exploited as enzymatic substrates or inhibitors [135,136]. The enzymatic biosensors of lipases are of great importance in analytical methods which find application in different sectors such as environmental science, the food industry, biodegradable polymers, and oleochemicals, and as diagnostic tools to detect the levels of cholesterol and triglycerides in blood samples [135]. The activity of lipases can be inhibited by different inhibitors. Some recent studies reported different inhibitors for lipases from various sources. For example, cetyltrimethyl ammonium bromide inhibits lipase from *Bacillus cereus* [137], Cadmium-II inhibits lipase from *Yarrowia lipolytica* [138] and cobalt-II inhibits lipase from *Mycobacterium tuberculosis* [139]. Moreover, lipase can also be used in electrochemical sensors and optical biosensors for the construction of different biosensors or bioassays [140].

## 6. Role of Lipases in Biodiesel Production

Lipases can be used in the esterification and transesterification of fats and oils to produce biodiesel (mono-fatty acid alkyl esters-FAAE) [90,141]. However, lipase application in biodiesel synthesis remains in its nascent stage, while its application in the industries mentioned above (food, leather, textile, etc.) is well established. In the transesterification reaction, when triglycerides are substrates in a nonaqueous media, the serine (Ser) nucleophile in the active center of the lipase participates in a charge relay system with histidine (His), glutamic acid (Glu), or aspartate (Asp) residues for the nucleophilic attack with alcohol on the ester bonds. In this reaction, an acetylated enzyme and an intermediate product are produced. After the formation of alkyl ester with a single carboxyl group in triacylglycerol, diacylglycerols are released as a byproduct. This is converted into monoacylglycerol, which releases the glycerol backbone in triacylglycerol and all stages of this reaction, and three mono fatty acid alkyl esters (biodiesel) are formed. This is then stabilized with more alcohol [142]. Over the last three decades, lipases have been used in the preparation of biodiesel from oils/fats. However, large-scale biodiesel production is catalyzed by different chemicals, such as acids and bases. The extensive application of lipases in biodiesel production is required to safeguard its position as a carbon-neutral fuel in the future.

Currently, some researchers use free lipase in biodiesel production [143,144]. However, most studies suggest the use of the immobilized form of lipase for this purpose [145]. Immobilized lipase plays a significant role in biodiesel production and various processes are optimized for this purpose. Microbial lipases are frequently used for biodiesel production and the choice of the lipases depends on their origin and formulation [58,144,145]. The important aspects of microbial lipases include their high activity, short reaction time, lower production inhibition, thermal tolerance, and the reusability of immobilized enzymes [146]. Microorganisms such as *Candida antarctica* [147], *Rhizopus arrhizus* [148], and *Bacillus aerius* [149] are used to produce lipases.

## 7. Role of Lipases in the Separation of Racemic Mixtures

The process in which pure enantiomers are converted into a racemic mixture is named racemization. Lipases are used to resolve the racemic mixture and to synthesize chiral building blocks for pharmaceutical and agrochemical compounds. Similarly, lipases can be used for the hydrolysis of water-insoluble esters and the resolution of stereoisomers by enantioselective hydrolysis [150]. As chirality plays an essential role in the efficacy of many medications, the synthesis of single enantiomers of pharmacological intermediates is a crucial process in drug manufacturing. Zhang et al. [151] described the immobilization of lipase from *Candida antarctica* (Novozym^®^ 435) for the kinetic resolution of racemic flurbiprofen using the method of enantioselective esterification with alcohols, as well as the procedure for optimizing the experimental resolution of the racemate, with a focus on solvent and alcohol types, inhibition of alcohol substrates, and the nature of the reversible reaction. Similarly, Baclofen is a gamma-aminobutyric acid (GABA) agonist that selectively activates the neurotransmitter GABA receptors to exert its antispasticity and analgesic effects. Muralidhar et al. [152] reported the use of baclofen (4-amino-3-(4-chlorophenyl) butanoic acid) in the treatment of pain and as a muscle relaxant. The isomers of baclofen have varied therapeutic effects depending on how they interact with receptors at the site of action. They discovered that *Candida cylindracea* lipase can be used as a catalyst to dissolve racemic combinations of a wide range of pharmaceutical substances, including baclofen.

## 8. Role of Lipases in Bioremediation

Our natural environment is constantly affected by xenobiotic compounds such as petroleum, insecticides, fertilizers, pesticides, plastics, and other hydrocarbon-containing substances. Several different approaches have been devised to eliminate these contaminants. However, these procedures are not effective or environmentally friendly. Compared to chemical and physical treatments, enzyme-based bioremediation is a mild process and is an easily adaptable method for eliminating these hazardous components from our natural ecosystem. However, production difficulties and the prohibitive costs of such enzymes limit their application in bioremediation. Hence, the use of microbial enzymes for bioremediation is gaining global importance. Moreover, microbial enzymes have a greater ability to transform contaminants into nontoxic substances and alleviate environmental pollution. Recently, Jacob et al. [153], isolated *Pseudomonas* sp. from petroleum oil-contaminated areas and tested it for lipase production. Lipase production was found to be high at temperatures ranging from 30 to 37 °C and at a pH range of 4 to 7. When compared to alternative carbon and nitrogen sources, the addition of sucrose and yeast extract to the medium increased enzyme synthesis. The authors concluded that *Pseudomonas* sp. can be employed to break down petroleum oil-polluted soils and should be considered a crucial component in formulating bioremediation strategies for petroleum oil spills. Several studies show that microorganisms that degrade hydrocarbons create lipolytic enzymes such as lipases. Blandon et al. [154] identified 56 lipolytic enzyme-producing bacteria in the deep-sea sediments of the Colombian Caribbean and discussed their bioremediation potential [154,155].

Similarly, the remediation of rising levels of micro-nano plastics (MNPs) in natural ecosystems is crucial, because MNPs adversely impact the health of living organisms in all ecosystems. Various researchers reported the use of plastic-resistant bacterial strains such as *Bacillus amyloliquefaciens* 1 and 2, and fungal strains such as *Aspergillus clavatus*, at waste disposal sites. All these microbes utilize different approaches to degrade plastics, either as a carbon source or because of the indirect action of microbial enzymes such as lipases. Bacterial and fungal species, including *Pseudomonas fluorescens*, *Pseudomonas aeruginosa*, and *Penicillium simplicissimum*, *Candida cylindracea*, *Rhizopus delemar,* are sources of lipases and are reported for their ability to degrade MNPs [156,157,158].

## 9. Recent Trends and Targets in Lipase Engineering

Lipases, with their high activity and stability, have a wide range of desirable applications. Researchers are focusing on the genetic modification and improvement of the scaling-up process for the efficient and economical production of lipases. Initially, the rational protein design (RD) approach was employed for enzyme engineering, which is based on the principles derived from the structural studies of enzymes and mechanistic evidence along with molecular biology techniques such as site-directed mutagenesis (SDM) [159,160]. The application of RD requires the exact identification of the residues responsible for the interaction of the substrate and the enzyme, and specificity, stability, and knowledge regarding the structure of the enzymes. RD is based on structural determination and structural–functional relationships, while molecular dynamics (MD) predicts possible mutations for the improvement of lipase enantioselectivity. Recently, MD and RD simulations are being used to improve the thermostability of lipases without reducing their activity [159,160]. The application of computational techniques and site-directed mutagenesis or the application of directed evolution (DE) techniques can be powerful tools for producing engineered lipases with optimized features (activity, stability, specificity, solubility, and optimum pH).

The directed evolution (DE) approach is based on error-prone polymerase chain reaction (ep-PCR) and DNA shuffling while it does not require the structural features of lipase [76]. In DE, randomly mutated products go through a DE process by imposing selection pressure and are then analyzed to reveal the enhanced features. The main steps in DE include random mutagenesis for library generation from a parent gene, insertion and expression in a competent host, and selection of the created mutant library. There are different targets in lipase engineering for improving overall characteristics, including thermostability, catalytic activity, and solvent tolerance [76,159].

### 9.1. Thermo-Stability

Lipases from thermophilic microorganisms, such as *Thermomyces lanuginose*, *Bacillus subtilis* and *Rhizopus orzae*, have been reported to have high (90 °C) heat tolerance [161,162,163]. However, the performance of the lipases is adversely affected by long-term temperature and hence cannot meet industrial standards. Rational design (RD) and directed evolution (DE) are used to enhance the thermostability of lipase. Wang et al., [164] cloned the *RCL* gene from *Rhizopus chinesis* and expressed it in *Pichia pastoris*. The lipase r27RCL had excellent results in the food and feed industry, but its use was limited due to its low thermostability. They used RD (FoldX) to calculate the free energy required to improve lipase thermostability, and a limited mutated pool of 19 residues containing 30 single-point mutations with reasonable free energy values was chosen from the 293 residues of lipase and subjected to gene expression, enzyme purification, and thermostability validation, resulting in a total of 4 mutations, identified as S142A, S250Y, Q239F, and D217V. The variation (m31) was created by integrating these 4 mutations and demonstrated a 5 °C higher optimum temperature than the wild type. The protein structure model was utilized to examine and explain molecular conformation changes, revealing that the increased hydrophobic stacking force inside particular secondary structures was primarily responsible for m31’s improved thermostability. They also performed MD simulations to investigate the mutant’s thermostability mechanism, and three of the four confirmed positive mutations were found in the thermally flexible areas. These findings strongly imply that using a combination of free energy-based RD methods and MD simulations to improve lipase thermostability would be a very beneficial strategy.

Yu et al. [165] reported an improvement in the thermostability of lipase from *Rhizopus chinensis* by two rounds of ep-PCR and two rounds of DNA shuffling in a DE approach. More recently, to increase the thermostability of lipase from *Pseudomonas Alcaligenes* (PaL). Similarly, Yu et al. [166] proposed a new technique named combined rational evaluation for thermostability engineering (CREATE). The hydrolysis of racemic menthol propionate to create L-menthol, one of the most used flavoring compounds in the food, cosmetic, and pharmaceutical industries, is catalyzed by the PaL lipase. However, L-menthol has low thermostability, which makes its use at higher temperatures difficult and limits its industrial applications. Three approaches were used in the CREATE strategy to forecast a pool of potential stabilizing mutations: sequence alignment with thermophilic orthologues, the Protein Repair One-Stop Shop (PROSS) application, and FireProt servers. Then, based on a reasonable evaluation of the placement in the 3D structure, free energy change, flexibility change, and distance from the active center, mutations with a high potential to improve stability were found. The researchers created 36 single-mutant variants and tested them for catalytic activity and thermostability, finding that 4 single-variants were more thermostable than wild-type PaL. The best 4M variant exhibited a 15-fold better half-life at 50 °C and a melting temperature (Tm) value higher by 14 °C compared to the wild type, and all feasible combinations of the 4 mutations were created for further improvement in stability. The CREATE method was found to be effective in guiding mutation selection and has the potential to be applied to other enzymes [166]. As multiple potential amino acid positions are implicated in probable mutations, many structural features play a role in lipase thermostability. In most cases, DE is a more effective technique for finding possible mutations than RD.

### 9.2. Catalytic Activity

Normally, lipases lack the desired features required for industrial processes because the maximum catalytic activity of natural lipases is observed in the 30–50 °C temperature range, whereas at higher temperatures, lipases exhibit a low reaction rate and reduced catalytic activity in the main process, resulting in a time-consuming and more expensive reaction. An improvement in the thermostability of the lipase enhances its tolerance to high working temperatures and is one of the main objectives of lipase engineering. Ma et al. [167] reported lipase of the thermophilic bacterium *Thermomicrobium roseum* DSM5159 (TrLip) for its enhanced thermostability and excellent solvent resistance. However, the catalytic activity of the lipase (TrLip) showed reduced activity against long-chain fatty acids, which greatly affects its utility in industrial processes. Hence, structural information and simulated natural evolutionary trends based on ancestral sequence reconstruction for lipase engineering were used, which significantly enhanced the catalytic activity and affinity against longer chains while maintaining the optimal pH and thermostability of the lipase. Thus, the enhanced catalytic activity and stability of TrLip can find applications in the food and chemical industries.

Using novel amino acid ionic liquids as chemical modifiers, Xu et al. [52] revealed theoretical and experimental evidence of the improved catalytic performance of lipase B from *Candida antarctica* obtained by chemical modification using amino acid ionic liquids. They concluded that the catalytic activity of modified *Candida antarctica* lipase B (CALB) was increased at various temperatures and pH, and that its thermostability and tolerance to organic solvents were improved. The modified CALB had improved structural stability and higher catalytic activity toward the substrate [52]. Furthermore, Tian et al. [168] presented a semi-rational DE approach combined with N-glycosylation for improving the methanol tolerance and catalytic activity of *Rhizomucor miehei* lipase in a single step for commercial biodiesel synthesis.

### 9.3. Solvent Tolerance

It has been demonstrated that the hydration state of proteins affects their physical characteristics as well as their reaction rates. However, it is unclear exactly how this impacts enzyme kinetics [169]. The importance of water activity in determining enzymatic activity has been recognized. In organic media, hydrolase-catalyzed esterification processes are frequently employed to measure enzyme activity. Lipases are frequently utilized in these types of reactions. The effect of water activity on the speed of reaction of lipases has been studied [170]. The results revealed that lipases from different sources react to an increase in water activity in different ways. The lipases from *Rhizopus arrhizus* exhibit optimum activity at low levels of water activity, while lipases from the *Pseudomonas* species exhibit activity that increases with water activity. Lipases from *Candida rugosa* have intermediate profiles with broader maxima [170,171].

Lipase deactivation can occur in synthetic reactions due to temperature changes, exposure to interfaces, and chemical denaturants, which are commonly present in esterification reaction systems as either substrates or products. Lipase deactivation can also occur due to physical changes in the enzyme structure or chemical changes such as disulfide bond breakage. Lipase activity is dependent on its tolerance to different solvent systems and its stability is a prerequisite for its application in various industrial processes. A lipase’s application potential may be evaluated under different solvent systems, and strategies are adopted to engineer lipase to tolerate the effects of organic solvents. The application of lipase in organic media exhibits various advantages such as increased activity, higher solubility of the substrate, and ease of downstream processing [172]. Tian et al. [168] reported N-glycosylation for the improvement of the methanol tolerance of *Rhizomucor miehei* lipase and revealed that the mutant N267 retained 64% activity after incubation in 50% methanol for 8 h. The formation of new hydrogen bonds resulted in this high methanol tolerance of N267 [168].

Methods such as genetic recombination, modification of amino acids, and immobilization onto support material are repeatedly employed for augmenting the thermostability and solvent tolerance of microbial lipases [19]. Recombinant lipases facilitate industrial processing, and the heterogeneous expression of lipase is considered a favorable method for enhancing the overall reaction. In the lipase genetic recombination strategy, the desired gene is cloned onto a vector and then inserted into the host for the heterologous expression of lipase with targeted characteristics. However, genetic modification does not guarantee the successful heterologous expression of the lipase gene [60,173]. For the recombinant heterologous expression of lipases, several microbial host strains such as bacteria, fungi, and yeast are used, and there are various expression vectors with significant variations in each host system for lipase production. *Escherichia coli* is the most commonly used prokaryotic host for heterologous lipase gene expression [174,175], and *Pichia pastoris* and *Saccharomyces cerevisiae* are the most commonly used eukaryotic host strains for lipase gene expression. Choosing the right strain is essential for optimum recombinant lipase yield [176].

Previous studies have shown that host selection necessitates the use of a suitable vector for lipase production. Each vector’s properties, notably its promoter, can influence the level of lipase yield. Immobilization techniques are employed to improve the lipase’s capacity to endure higher temperatures in the presence of organic solvents in various industrial processes because immobilization provides mechanical stability and allows the lipase to be reused [60,173]. Furthermore, the high cost of producing thermostable and solvent-tolerant lipases remains a key barrier in industrial applications. Researchers are focusing on using protein engineering approaches to establish a viable low-cost manufacturing process for these lipases. Thermostability and solvent tolerability will increase the durability and reusability of lipases in many industrial processes. Similarly, a robust immobilization mechanism will also improve these characteristics.

## 10. Conclusions and Future Prospects

The current review shows that lipases are versatile biocatalysts used in various bioconversion reactions. Microbial lipases have a wide range of applications and are produced in significant quantities across industries. Most lipases screened from microorganisms are secretory extracellular enzymes that can be isolated with high purity and are suitable for mass production as required by various industries. In recent years, researchers have focused on microbial lipases from novel microbes. With the rapid development of enzyme technology and its pragmatic application in various industrial processes, these enzymes are receiving a great deal of attention. There have also been major developments in the areas of microbial lipase production from different microbial sources and purification. The present review provides updated information on lipase production from different microbial sources, and from classical and modern purification techniques, including precipitation and chromatographic separation, the immunopurification technique, the reversed micellar system, ATPS, and ATPF, and the use of microbial lipases in different industries, viz., the food, textiles, leather, cosmetics, paper, and detergent industries. Moreover, the present review illustrates the use of lipases in biosensors, biodiesel production, and tea processing, and lipases’ role in bioremediation, with a special emphasis on the lipase engineering targets of enhanced specificity and thermostability.

Lipase characteristics have greatly improved because of genetic engineering and protein immobilization techniques. Future trends for lipase immobilization include immobilization on hydrophobic surfaces via interfacial activation. Maintaining lipase in its active form and hyperactivating lipase for industrial applications have proven to be quite beneficial. Future studies will focus primarily on optimizing lipase-catalyzed reactions techniques in consideration of actual production. Over time, the demand for microbial lipases for use in different industries is increasing due to their capability of prominently enhancing various biotechnology-based production processes. Researchers are focusing on screening and generating novel lipases with unique characteristics by searching various microbial sources, using bioinformatics tools, and with the application of recombinant DNA technology. Novel approaches would be employed to improve the properties of existing microbial lipases, such as specificity, productivity, and thermostability. Furthermore, the commercial applicability of lipases can be improved by adopting novel research focused on the source, biochemistry, enzymology, and application of lipases in different industrial processes.

## Figures and Tables

**Figure 1 microorganisms-11-00510-f001:**
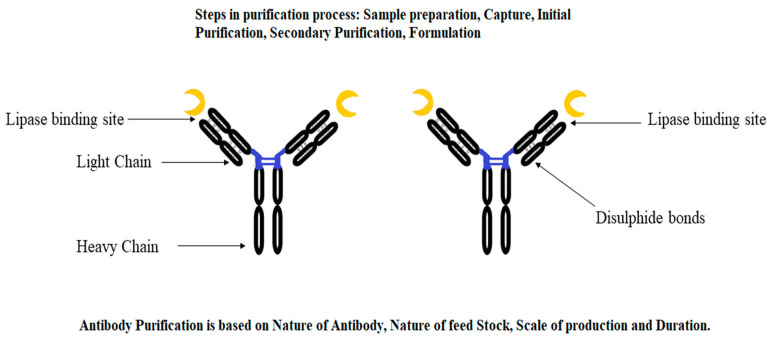
A representative diagram of an antibody used for the immunopurification of microbial lipases.

**Figure 2 microorganisms-11-00510-f002:**
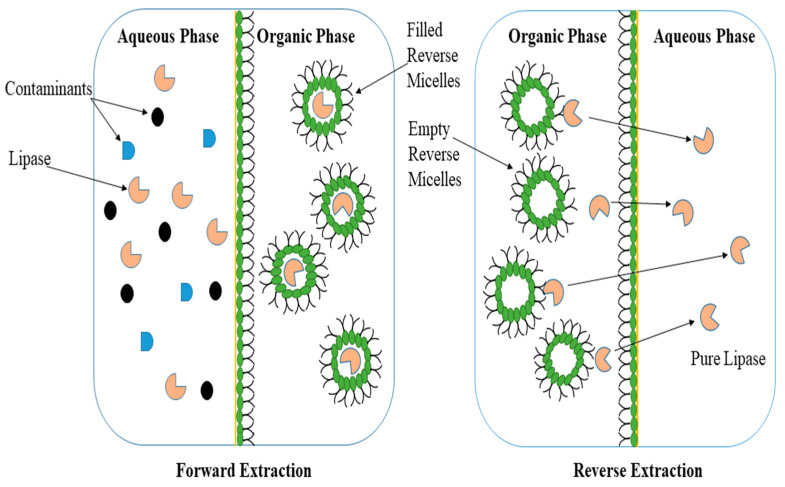
A schematic diagram of the two-step extraction in/with micellar systems.

**Figure 3 microorganisms-11-00510-f003:**
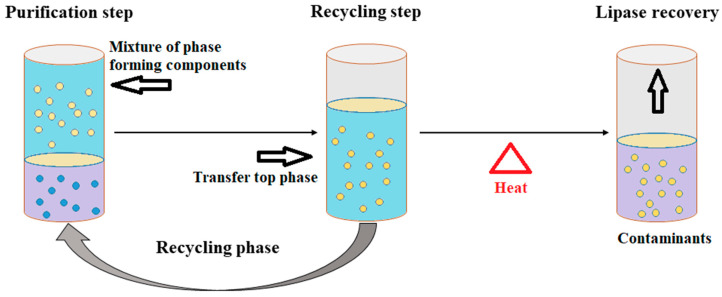
Schematic presentation of an aqueous two-phase system (ATPS) for lipase purification.

**Figure 4 microorganisms-11-00510-f004:**
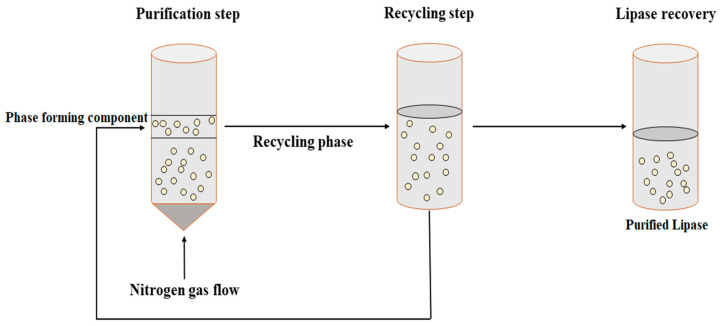
Schematic presentation of aqueous two-phase flotation (ATPF) for lipase purification.

## Data Availability

Data sharing not applicable, No new data were created or analyzed in this study. Data sharing is not applicable to this article.

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
