# Peer review of "The Recent Advances in the Utility of Microbial Lipases: A Review"

_microorganisms, 2023, doi:10.3390/microorganisms11020510_

Round 1

Reviewer 1 Report

The manuscript nicely describes the most recent advances in lipases development and application, from their production to their industrial uses. All aspects neccessary for industrial application in different áreas are well described. Only, a few relevant points should be addressed before its publication.

In general, the manuscript is well written, but still the english gramar should be revised along the paper. Some examples are: use of infinive verbes tense instead of their past tense, in many cases the comparative term “compared to” should be better replaced by “compared with”, etc.

Importantly, the aspects treated in each section are based on a few literature references of the issues described (2-4 refs/section). This is a small number of refs to be described in a “comprehensive review”. Many of the most relevant publications with relevant contributions in each of these sections are not being cited/considered. Instead, a few literatura examples are being described as literature examples.  Hence, to my eyes the manuscript is rather “an overview” of the present situation of lipases use & development, and then the Words “comprenhensive review” should not appear in the manuscript title. Instead,

“Overview” or “the most recent advances in” should be more appropriate for the title of this manuscript.

I also suggest to include some of the most recent reviews published on different aspects of lipase catalysts.

The section describing the lipases resistance to solvent media does not properly reflect the right origin of the poor lipases stability in some organic solvents (specially the polar ones). This is relevant to any biocatalytic process in low wáter activity media. I recommend to authors to include some of the most relevant refs describing the effect of wáter activity on both, lipases activity and stability in non-conventional media.

Other aspects to be corrected are indicated in the attached file with the corrected versión of the manuscript

Author Response

Comments and Suggestions for Authors

Comment: The manuscript nicely describes the most recent advances in lipases development and application, from their production to their industrial uses. All aspects neccessary for industrial application in different áreas are well described. Only, a few relevant points should be addressed before its publication.

In general, the manuscript is well written, but still the english gramar should be revised along the paper. Some examples are: use of infinive verbes tense instead of their past tense, in many cases the comparative term “compared to” should be better replaced by “compared with”, etc.

Author’s response: We are thankful to you for your positive comments. We have revised the manuscript thoroughly and incorporated all suggested changes. Moreover, we have done English editing by English-Language Paper Editing Service “NURISCO” (certificate attached: (YNU-221216-01) and the manuscript is submitted for your kind consideration, please. Thank you.

Comment: Importantly, the aspects treated in each section are based on a few literature references of the issues described (2-4 refs/section). This is a small number of refs to be described in a “comprehensive review”. Many of the most relevant publications with relevant contributions in each of these sections are not being cited/considered. Instead, a few literatura examples are being described as literature examples.  Hence, to my eyes the manuscript is rather “an overview” of the present situation of lipases use & development, and then the Words “comprenhensive review” should not appear in the manuscript title. Instead, “Overview” or “the most recent advances in” should be more appropriate for the title of this manuscript.

Author’s response: We completely agree with your kind comments. We have cited some recent articles in each section and revised the title accordingly.

Comment: I also suggest to include some of the most recent reviews published on different aspects of lipase catalysts.

Author’s response: We have cited some of the most recent reviews published on different aspects of lipase catalysis. Thank you.

Comment: The section describing the lipases resistance to solvent media does not properly reflect the right origin of the poor lipases stability in some organic solvents (specially the polar ones). This is relevant to any biocatalytic process in low wáter activity media. I recommend to authors to include some of the most relevant refs describing the effect of wáter activity on both, lipases activity and stability in non-conventional media.

Author’s response: We have added some of the most relevant references describing the effect of water activity on both, lipase activity and stability in non-conventional media. Thank you.

Comment: Other aspects to be corrected are indicated in the attached file with the corrected versión of the manuscript

Author’s response: We have added all suggested changes in the revised manuscript. Thank you.

Reviewer 2 Report

Manuscript ID: microorganisms-2056284

Title: Utility of Microbial Lipases: A comprehensive review

Authors

Lipases are important enzymes for many production processes. Recently microbial-produced lipases have taken relevance worldwide and have a great market, novel applications of lipases have been reviewed in the manuscript. The theme of the proposed manuscript has scientific relevance, but in the actual state, the manuscript is not suitable for publication.

The main weakness identified in the manuscript are summarized and described below

Overall, the redaction in the manuscript must be deeply reviewed, there are several mistakes all over the document.

The manuscript aims “to deliver a critical analysis of lipase-producing microbes, distinguished from the previously published reviews and illustrates the use of lipases in biosensors, biodiesel production, tea processing, and their role in bioremediation and racemization”, but the aim of the manuscript is not well addressed, the manuscript does not describe in detail such mentioned gaps on previously published reviews.

The in-text references in many sections of the manuscript must be reordered alphabetically and according to the year of publication.

The authors describe in the manuscript that “recent studies reported microorganisms and their potential to produce lipases are mentioned in Table 1”, more over the title of table 1 is “Lipases from different microbial sources and their applications”, but just a list of different microorganisms is shown in table 1, without mention of the lipases produced by them or their applications. 

There are many formatting errors in Table 1, as well as different formats in the references, in addition to the fact that the information presented does not allow establishing the potential of the microorganisms referred to as producers of lipases with potential for applications in different biotechnological processes.

In the manuscript, there are several paragraphs with minimal or no references to support the information described.

All figures need to be improved, they are not clear, for example in figure 1, a cartoon of an IgG antibody is shown, but this antibody format has multiple applications, not only for protein purification, figure 1 must include more information about the lipase purification technique. In figures 3 and 4, the phases employed are not illustrated in the figures, is not clear how lipases are purified in the systems.

In the section of modern purification techniques, just one study is described as an example of the application of each modern lipase purification technique, more studies must be included in the mentioned section, and the descriptions of the techniques must be improved.

In the reverse micellar systems section, surfactants are mentioned, however, examples, nature, sources, and characteristics of such surfactants not were described.

Author Response

Comment: Lipases are important enzymes for many production processes. Recently microbial-produced lipases have taken relevance worldwide and have a great market, novel applications of lipases have been reviewed in the manuscript. The theme of the proposed manuscript has scientific relevance, but in the actual state, the manuscript is not suitable for publication. The main weakness identified in the manuscript are summarized and described below

Overall, the redaction in the manuscript must be deeply reviewed, there are several mistakes all over the document.

The manuscript aims “to deliver a critical analysis of lipase-producing microbes, distinguished from the previously published reviews and illustrates the use of lipases in biosensors, biodiesel production, tea processing, and their role in bioremediation and racemization”, but the aim of the manuscript is not well addressed, the manuscript does not describe in detail such mentioned gaps on previously published reviews.

Author’s response: We are thankful to you for your comments. We have revised the manuscript thoroughly and addressed all suggested changes by adding more information based on the aims and objectives section and cited recent published articles on lipases. Thank you.

Comment: The in-text references in many sections of the manuscript must be reordered alphabetically and according to the year of publication.

Author’s response: All references were reordered and rectified. Thank you.

Comment: The authors describe in the manuscript that “recent studies reported microorganisms and their potential to produce lipases are mentioned in Table 1”, more over the title of table 1 is “Lipases from different microbial sources and their applications”, but just a list of different microorganisms is shown in table 1, without mention of the lipases produced by them or their applications. There are many formatting errors in Table 1, as well as different formats in the references, in addition to the fact that the information presented does not allow establishing the potential of the microorganisms referred to as producers of lipases with potential for applications in different biotechnological processes.

 Author’s response: We have revised table 1 thoroughly and added the suggested changes. Thank you

Comment: In the manuscript, there are several paragraphs with minimal or no references to support the information described.

Author’s response: We have cited relevant references in the required paragraphs. Thank you

Comment:  All figures need to be improved, they are not clear, for example in figure 1, a cartoon of an IgG antibody is shown, but this antibody format has multiple applications, not only for protein purification, figure 1 must include more information about the lipase purification technique. In figures 3 and 4, the phases employed are not illustrated in the figures, is not clear how lipases are purified in the systems.

Author’s response: We have improved all figures and added more details in the revised manuscript. Thank you.

Comment: In the section of modern purification techniques, just one study is described as an example of the application of each modern lipase purification technique, more studies must be included in the mentioned section, and the descriptions of the techniques must be improved.

Author’s response: We are thankful to you for your kind comments. We have added more details in the section on modern purification techniques. Thank you

Comment: In the reverse micellar systems section, surfactants are mentioned, however, examples, nature, sources, and characteristics of such surfactants not were described.

Author’s response: We have added the required information in the section of the micellar system. Thank you

Reviewer 3 Report

The present contribution by Ali et al. presents the general information on lipases from microbial sources, purification methods and their use in various industries. 

I think the present Review is clear and generally well-structured. As an overall consideration, sub-chapters are suggested, for improving the manuscript structure (like it was implemented for “Purification strategies of microbial lipases” paragraph).

The manuscript was well written; however, this review did not give any interesting future perspectives apart from concluding with already established facts in the field. The manuscript would gain in value if added a separate section of future direction and perspectives.

In the introduction, it would be appropriate to indicate what distinguishes the present work in comparison to other recently published literature reviews, such as S. Mahboob, K. Tahir, S. Ali; A SYSTEMATIC OVERVIEW ON THE UPSTREAMING, DOWNSTREAMING AND INDUSTRIAL APPLICATIONS OF MICROBIAL LIPASES; INT. J. BIOL. BIOTECH., 19 (2): 171-182, 2022.

In some paragraphs also lack recent information, for example in paragraph “(B) Modern purification techniques”, the literature on immune-purification is from 2004 and from reverse micellar systems from 2012.

In Table 1, most of the data is from 2020, although an entry from 2029 can be found. As 2022 is already coming to an end, some of this year's reports could be included.

I propose that a short paragraph on the use of lipases in the tea processing be included as part of the food industry paragraph.

The paragraph “Role of lipases in racemization” describes examples of separation of racemic mixtures and not of racemisation processes, therefore I propose to rename this section. It would also be of added value to include schemes of the kinetic resolutions, taking into account the stereoselectivity of the reaction.

I also suggest some others minor corrections:

Extra spaces or missing spaces (if the lines in the text were numbered I would state exactly which ones I meant)

The use of italics or their absence where they should be used: (Table 1), page 16 Mycobacterium tuberculosis

Author Response

Comment: The present contribution by Ali et al. presents the general information on lipases from microbial sources, purification methods and their use in various industries. 

I think the present Review is clear and generally well-structured. As an overall consideration, sub-chapters are suggested, for improving the manuscript structure (like it was implemented for “Purification strategies of microbial lipases” paragraph).

The manuscript was well written; however, this review did not give any interesting future perspectives apart from concluding with already established facts in the field. The manuscript would gain in value if added a separate section of future direction and perspectives.

Author’s response: We are thankful to you for your positive comments. We have added more information in future research directions. Thank you

Comment: In the introduction, it would be appropriate to indicate what distinguishes the present work in comparison to other recently published literature reviews, such as S. Mahboob, K. Tahir, S. Ali; A SYSTEMATIC OVERVIEW ON THE UPSTREAMING, DOWNSTREAMING AND INDUSTRIAL APPLICATIONS OF MICROBIAL LIPASES; INT. J. BIOL. BIOTECH., 19 (2): 171-182, 2022.

Author’s response: We have added the novelty of the present work in the introduction section in comparison with already published work. Thank you 

Comment: In some paragraphs also lack recent information, for example in paragraph “(B) Modern purification techniques”, the literature on immune-purification is from 2004 and from reverse micellar systems from 2012.

Author’s response: We have cited recent studies in the said section. Thank you

Comment: In Table 1, most of the data is from 2020, although an entry from 2029 can be found. As 2022 is already coming to an end, some of this year's reports could be included.

Author’s response: We are thankful to you for your kind comments. We have removed 2029 and added some recent reports from 2022. Thank you

Comment: I propose that a short paragraph on the use of lipases in the tea processing be included as part of the food industry paragraph.

Author’s response: We have added the said paragraph in the food industry section. Thank you

Comment: The paragraph “Role of lipases in racemization” describes examples of separation of racemic mixtures and not of racemisation processes, therefore I propose to rename this section.

Author’s response: The section was renamed as “Role of lipases in the separation of racemic mixtures”. Thank you.

Comment: I also suggest some others minor corrections:

Extra spaces or missing spaces (if the lines in the text were numbered I would state exactly which ones I meant).

The use of italics or their absence where they should be used: (Table 1), page 16 Mycobacterium tuberculosis

Author’s response: We have incorporated the suggested changes in the revised manuscript. Thank you

Round 2

Reviewer 2 Report

Authors

The authors have considerably improved the manuscript quality, however, still have pending several of the previously indicated comments, major corrections must be addressed in the manuscript, especially in table 1 and the figures.

In Table 1. Microbial sources of lipases from diverse environments, the descriptions of the microorganisms are too general, and not focused on the lipase production potential of such microorganisms, the name of the table indicates that the microorganisms included are from different environments, but information about the environment from the microorganisms were isolated is not included in the table. The references in table 1 are not in adequate format. Table 1 needs to include more information about the lipase production potential of the included microorganisms and the environments from which they were isolated, the format of the reference must be reviewed and corrected.

All figures still need to be improved, they are not clear, for example in figure 1, a cartoon of an IgG antibody is shown, but this antibody format has multiple applications, not only for protein purification, figure 1 must include more information about how antibodies are employed in lipase-purification. In figures 3 and 4, the different phases employed in the purification techniques are not illustrated, and it is not clear how lipases are purified in the systems. The figures included in the manuscript must be clear and understandable by themselves inclusive for not specialized readers

Author Response

Response to reviewer’s comments: We are thankful to you for your positive response and pragmatic suggestions. We have incorporated all suggested changes in table 1 and revised the figures by adding relevant information accordingly.

Reviewer 3 Report

The work has been significantly improved. In the current version I only have a comment to the chemical names as on page 18 (I suggest changing y-aminobutyric acid to γ-aminobutyric acid or gamma-aminobutyric acid or even better 4-Aminobutanoic acid and be consistent in the rest of the chemical names used in the manuscript). Correct chemical name for baclofen is 4-amino-3-(4-chlorophenyl)butanoic acid. On page 23, I suggest you correct “reac-tions”.

Author Response

Response to reviewer’s comments: We are thankful to you for your kind suggestions. We have incorporated all suggested changes accordingly.